# PD-1/PD-L1 Inhibitors Response in Triple-Negative Breast Cancer: Can Long Noncoding RNAs Be Associated?

**DOI:** 10.3390/cancers15194682

**Published:** 2023-09-22

**Authors:** Carolina Mathias, Vanessa Nascimento Kozak, Jessica Maria Magno, Suelen Cristina Soares Baal, Victor Henrique Apolonio dos Santos, Enilze Maria de Souza Fonseca Ribeiro, Daniela Fiori Gradia, Mauro Antonio Alves Castro, Jaqueline Carvalho de Oliveira

**Affiliations:** 1Post-Graduation Program in Genetics, Department of Genetics, Federal University of Parana, Curitiba 81530-980, Brazil; carol.mathias1@hotmail.com (C.M.);; 2Post-Graduation Program in Bioinformatics, Bioinformatics and Systems Biology Laboratory, Federal University of Paraná, Curitiba 81520-260, Brazilvictor.henrique.apolonio@gmail.com (V.H.A.d.S.);

**Keywords:** immune response, tumor microenvironment, immunotherapy, HCP5, UCA1

## Abstract

**Simple Summary:**

The recent approval of the drug Pembrolizumab for patients with the triple-negative subtype of breast cancer highlights the search for new markers to indicate the treatment. Due to their broad regulatory role and tissue specificity, lncRNA molecules can rise as biomarkers. Therefore, this review aimed to raise possible lncRNAs that act as an indication for the treatment. In addition, we searched for lncRNAs already discussed in other types of cancer that deserve to be better investigated in the context of breast cancer.

**Abstract:**

As immune checkpoint inhibitors (ICI) emerge as a paradigm-shifting treatment option for patients with advanced or metastatic cancer, there is a growing demand for biomarkers that can distinguish which patients are likely to benefit. In the case of triple-negative breast cancer (TNBC), characterized by a lack of therapeutic targets, pembrolizumab approval for high-risk early-stage disease occurred regardless of PD-L1 status, which keeps the condition in a biomarker limbus. In this review, we highlight the participation of long non-coding RNAs (lncRNAs) in the regulation of the PD-1/PD-L1 pathway, as well as in the definition of prognostic immune-related signatures in many types of tumors, aiming to shed light on molecules that deserve further investigation for a potential role as biomarkers. We also conducted a bioinformatic analysis to investigate lncRNAs already investigated in PD-1/PDL-1 pathways in other cancer types, considering the TNBC molecular context. In this sense, from the generated data, we evidence here two lncRNAs, UCA1 and HCP5, which have not yet been identified in the context of the tumoral immune response in breast cancer. These candidates can be further explored to verify their use as biomarkers for ICI response. In this article, we present an updated review regarding the use of lncRNA as biomarkers of response to ICI, highlighting the versatility of using these molecules.

## 1. Introduction

Since November 2020, the Food and Drug Administration (FDA) granted regular approval to pembrolizumab, an anti-programmed cell death protein 1 (PD-1) monoclonal antibody, in combination with chemotherapy for patients with locally recurrent unresectable or metastatic triple-negative breast cancer (TNBC) whose tumors express programmed-death ligand 1 (PD-L1) with a combined positive score (CPS) of more than 10 [1]. In July 2021, the drug was approved for another indication: high-risk, early-stage TNBC in the neoadjuvant and adjuvant settings [2]. The approval was based on the results of the phase-III KEYNOTE-522 trial (NCT 03036488), which enrolled patients regardless of PD-L1 expression and found higher rates of pathological complete response among patients receiving pembrolizumab in addition to neoadjuvant chemotherapy. This benefit was consistent across all subgroups in this trial, different from the IMpassion130 trial (NCT02425891) studying atezolizumab and nab-paclitaxel in TNBC, which observed clinical benefit only in patients whose tumors expressed PD-L1 [3,4]. Those conflicting results highlight a common unmet need in the clinical setting: biomarkers capable of discriminating against patients more likely to respond to a drug. A comprehensive understanding of TNBC’s molecular context and tumor microenvironment is the pathway to elucidate susceptibility—and resistance—to immunotherapy.

TNBC is a heterogeneous disease, and based on this, Lehmann et al. proposed a classification of this subtype into six additional groups, which exhibit distinct pathways’ dysregulation and prognosis [5]. This TNBC classification is based on messenger RNA (mRNA) expression; however, protein-coding gene sequences account for less than 2% of the human genome sequence. So, most transcribed parts of the genome, the non-coding RNAs (ncRNAs) sequences, were underexplored as potential markers [6]. Among ncRNAs, the long non-coding (lncRNAs) are transcripts with more than 200 nucleotides, which have no or small coding ability and show features similar to mRNAs, such as polyadenylation and complex splicing patterns. Its genes are often transcribed by RNA polymerase II [7]. LncRNAs’ use as biomarkers has been investigated in TNBC development, progression, and therapy response [8,9,10,11]. They have also been identified to regulate the PD-1/PD-L1 pathway, leading to participation in the immune response and, consequently, immunotherapy response [12].

In this context, we herein review information regarding lncRNAs described in the PD-1/PD-L1 regulatory network to highlight candidate lncRNAs that may act as biomarkers for stratification of TNBC patients and aid in refining patient selection for treatment with pembrolizumab and other immunotherapies. We also investigated the role of lncRNAs already discussed in different cancer immune contexts using TNBC data.

## 2. TNBC Immune Subtypes and Tumor Microenvironment (TME)

TNBC is the most aggressive breast cancer subtype, and early relapse and absence of therapeutic targets are significant problems in its treatment [13]. Recent research demonstrated that some TNBCs have higher immunogenicity than other breast cancer subtypes [14]. Thus, immunotherapy can be evaluated as an option for some patients [5,15,16,17,18,19].

The so-called “hot tumors”, also classified as immunoreactive, have a better prognosis and a more significant response to immunotherapy. In comparison, “cold tumors” have worse prognosis and poor response to immunotherapy [20]. Immunoreactive tumor microenvironments are composed mainly of natural killer (NK) cells, a type of cytotoxic lymphocyte that are crucial components of the innate immune response; T CD8+, which are the main kind of cytotoxic lymphocytes in tumors; and M1 macrophages, also called classical macrophages, which are pro-inflammatory and can activate the immune response and oppose tumorigenesis. In contrast, immunosuppressive tumor microenvironments are formed by M2 macrophages, forkhead box P3+ (Foxp3+) regulatory T lymphocytes (Tregs), and myeloid-derived suppressor cells (MDSCs) [21].

Based on TNBC tumor microenvironment differences, large amounts of data were generated to better characterize it, including intrinsic subtypes according to tumor immune response. Table 1 summarizes data from studies that aimed to classify TNBC based on immune profile using different methodologies. These data highlight the heterogeneity observed in TNBC and its complexity regarding the immune response.

Thus, it is clear that TNBC is heterogeneous in terms of the composition of immune cells forming the TME. This heterogeneity is being further elucidated, especially with regard to the PD-1/PD-L1 pathway. In addition, we also highlight the important role that regulatory molecules can play in this pathway. In this work, we will focus on the regulatory role of lncRNAs in order to bring to light potential new biomarkers that can be used as allies in the indication of immunotherapy.

## 3. LncRNAs and Immune Response: The PD-1/PD-L1 Pathway

LncRNAs are recently gaining more attention in cancer research due to their high versatility in regulating gene expression. Many studies have already shown the role of lncRNAs in cancer hallmarks, and over the past years, the literature on the relation between lncRNAs and tumor immune response has grown significantly [27,28,29,30,31].

In general, the cancer-immunity cycle can be divided into seven steps, and lncRNAs are known to participate in all of them: (1) cancer cells release antigens; (2) antigens are captured by dendritic cells (DCs), (3) DCs with captured antigens migrate to lymph node and prime with T cells to activate tumor-specific cytotoxic CD8+ T cells, (4) the cytotoxic T cells migrate from lymph node into blood vessels, (5) immune cells infiltrate into tumor stroma, (6) the cytotoxic T cells recognize tumor cells and (7) T cells kill cancer cells [32]. Tumor immune escape is considered the last phase of the cancer immunoediting process. In this event, tumor cells acquire the ability to avoid immune system recognition and interrupt the aforementioned cycle. Immune evasion is one of the cancer hallmarks that can be targeted for treatment, including the PD-1/PD-L1 pathway, where lncRNAs have also been previously described as acting in such biological processes [28].

PD-1, also known as CD279, is a 288 amino acids transmembrane protein, which is a member of the immunoglobulin superfamily related to CD28 and to cytotoxic T-lymphocyte-associated protein 4 (CTLA-4) [33,34]. It is expressed on the surface of different immune cells, including lymphocytes, macrophages, dendritic cells, and monocytes [35,36]. While PD-1 is a CD28 family member, its ligands—PD-L1 (or B7-H1) and PD-L2 (or B7-DC)—are part of the B7 family of immune-regulatory ligands. These latter are cell-surface protein ligands that bind to the CD28 family of receptors on lymphocytes to mediate immune response through co-stimulatory and co-inhibitory signals [34,37]. In that context, the PD-1/PD-L1 axis has a co-inhibitory function and has a physiological role in maintaining immune homeostasis—by inhibiting a prompt immune response, it protects the host against autoimmunity [35]. However, some tumors take advantage of this pathway to “hide” from the immune response by overexpressing PD-L1 [38]. Monoclonal antibodies (mAbs) targeting this pathway can activate anti-tumor immunity and are increasingly used in clinical practice. The first approved immune checkpoint inhibitors (ICI) targeting this specific pathway included pembrolizumab and nivolumab, which are anti-PD-1 mAbs, followed by atezolizumab, avelumab and durvalumab, which are anti-PD-L1 mAbs [39]. Within this context, the PD-L1 expression on tumor surface is currently used as one of the biomarkers to assess eligibility for immunotherapy, along with microsatellite instability (MSI) and tumor mutational burden (TMB)—of note, all these markers are rarely concordant, but each of them suffices as a marker [40]. Nevertheless, the variable cutoffs to predict response among different tumor sites and the large range of variation in tumor response for those selected based on PD-L1 expression show that ligand expression alone is not a robust biomarker [41]. Supra-regulatory mechanisms of the PD-1/PD-L1 axis, including those involving lncRNAs, could respond to part of this variation and, thus could be explored as biomarkers [28].

In breast cancer, PD-L1 expression is up-regulated in tumor tissue compared to the normal counterpart. Besides that, when tumors were stratified according to subtypes, TNBC exhibited significant upregulation of PD-L1 expression compared with patients with luminal subtypes [42]. The regulatory roles of lncRNAs in the PD-1/PD-L1 pathway have been extensively explored in recent years.

Below, we gather information about these regulatory mechanisms on breast cancer, especially on TNBC, and also in other types of tumors, which were then analyzed considering the BC context.

### 3.1. lncRNAs and PD-1/PDL-1 Axis in Breast Cancer

The lncRNAs MALAT1 and XIST are two important cancer-related molecules previously studied in several breast cancer contexts [43,44]. The participation of these lncRNAs in the PD-1/PD-L1 pathway was explored by Samir and colleagues. In this study, the authors introduced a novel immune-modulatory competing endogenous RNA (ceRNA) network in BC, the MALAT1-XIST/miR-182-5p/PD-L1 axis. CeRNA network usually includes a post-transcriptional regulation of a protein expression caused by an RNA molecule (lncRNA, pseudogene, etc.) binding competitively with miRNA response elements and forming a complex regulatory network. Using the TNBC cell line MDA-MB-231, it was demonstrated that miR-182-5p could act as an oncomiR by promoting the upregulation of oncogenic PD-L1 and the lncRNA MALAT1, in addition to downregulating the expression of the tumor suppressor gene XIST in the same cells. Because of the role of MALAT1/XIST in regulating PD-L1 expression, the authors suggest those as possible immunotherapeutic targets [42].

XIST was also evaluated in another ceRNA axis, the miR-194/XIST/PD-L1, using MDA-MB-231 cell lines [45]. Its expression was inversely correlated with PD-L1 expression levels in TNBC cell lines, shedding light on XIST’s role as a potential BC immune biomarker. Another study investigated the relation between TSIX and PD-L1. It was verified that TSIX silencing induced PD-L1 repression, suggesting a positive correlation between them [46].

The lncRNA T-cell leukemia/lymphoma 6 (TCL6) expression was also positively correlated with tumor lymphocytic infiltration and PD-1, PD-L1, PD-L2 and CTLA-4 expression [47]. Another lncRNA evaluated in TNBC participating in the PD-1/PD-L1 axis was GATA3-AS1, which is highly expressed in TNBC tissues and has an oncogenic role. Besides promoting immune evasion through an increased PD-L1 expression, GATA3-AS1 decreases GATA3 expression to induce immune evasion by regulating the miR-676-3p/COPS5 pathway [48].

The tissue differentiation-inducing non-protein coding RNA (TINCR), a lncRNA known to act as an oncogene in several types of tumors, including breast cancer, was investigated by Wang et al. regarding its role in tumor immunity [12]. The expression of this lncRNA is able to increase the expression of PD-L1 by recruiting DNMT1, which promotes methylation of miR-199a-5p, downregulating its expression. TINCR also sponges miR-199a-5p, which favors the stability of its target mRNA USP20. USP20 overexpression inhibits PD-L1 ubiquitination. In breast cancer, there is also an upstream regulatory mechanism in which the INFγ–STAT1 signaling axis promotes TINCR transcription in the breast. Thus, the STAT1-TINCR-USP20-PD-L1 upregulates PD-L1 expression and promotes tumor progression in vitro and in vivo. Besides this, TINCR knockdown enhanced sensitivity to PD-L1 inhibitor sensitivity in an in vivo model of breast cancer [12].

LncRNA LINK-A overexpression was verified to be related to decreased infiltration by CD8+ T cells and antigen-presenting cells (APC) in breast cancer, suggesting impaired antigen presentation and, thus decreased sensitivity to PD-1 blockade [49]. LINK-A exerts an oncogenic effect through a signaling pathway that results in the degradation of the antigen peptide-loading complex (PLC) and of the tumor suppressors Rb and p53. Hu et al. demonstrated that counteracting the LINK-A cascade resulted in PLC, Rb and p53 stabilization, which sensitized mammary gland tumors to ICI.

When evaluating clinical and genetic data from 1099 patients with breast cancer available in The Cancer Genome Atlas (TCGA) database, Tan et al. reported a correlation between eight lncRNAs related to pyroptosis and their impact on the expression of other lncRNAs, as well as the potential application of these genes as biomarkers for prognosis and immune status in breast cancer patients. Among the lncRNAs studied, three showed a positive correlation with survival and were considered sensitive to PD-L1 inhibitors [50].

As shown in Table 1, several studies correlate immune subtypes with prognosis, not only in breast cancer but in many different types of cancer. In this context, there are also many studies correlating immune-related lncRNA signatures with prognosis, and those signatures can be further explored to be extrapolated as predictive markers to respond to immunotherapy eventually. Four recently published studies [51,52,53,54] proposed signatures composed of different immune-related lncRNAs as predictive of prognosis and immune infiltration in breast cancer. We organized the data from each study in Table 2. The lncRNAs included in each study are listed in Table 2. According to these data, only AL161646 was found in more than one of these signatures, showing a long way to go to a possible definition of relevant markers, standardization, and validation, which is the path to implementation in clinical practice. Of note, LINC01871, cited in one of the signatures as a protective factor [51], was also related to immune response activation and favorable overall survival in basal-like samples in a study from our group [55].

In Figure 1 we organized the data of lncRNAs and the PD-1/PDL-1 pathway.

### 3.2. lncRNAs and PD-1/PDL-1 Axis in Other Cancer Types

Metastatic melanoma was the first tumor for which treatment with an immune checkpoint inhibitor was approved [56]. Though patients with advanced unresectable melanoma are systematically considered for immunotherapy, with objective response rates around 40% in the first-line setting [57], there is no current biomarker used for patient selection yet. The National Comprehensive Cancer Network (NCCN) guidelines for melanoma state “Testing for tumor PD-L1 should not guide clinical decision-making. The utility of this biomarker requires further investigation” [58]. This unfilled niche for a biomarker for immunotherapy response in melanoma is an urgent issue, as the lack of it can translate into increased financial expenses and unnecessary exposure to toxicities [59]. Trying to fill this gap, Li et al. identified a lncRNA—lncRNA inducing MHC-I and immunogenicity of tumor or LIMIT—which is responsive to IFNγ in both human melanoma and mouse cells and induces MHC-I and MHC-II expression, being able to enhance immunotherapy efficacy [30]. Zhou et al. also described a 15 lncRNAs signature distinguishing melanoma patients more likely to respond to anti-PD-1 monotherapy [60].

In bladder cancer, Zhou et al. also described a lncRNAs signature of tumor-infiltrating B lymphocytes associated with an immunosuppressive phenotype [61], which could be used to predict immunotherapy response.

Yu et al. used information from melanoma and bladder cancer samples, identifying lncRNA-based immune subtypes associated with survival and response to cancer immunotherapy. Based on this study, four distinct classes were identified, namely: immune-active class, immune-exclusion class, immune-dysfunctional class, and immune-desert class. The immune-active class showed the most clinical benefit with immunotherapy [62].

Colorectal cancer is always a subject when it comes to immunotherapy. Microsatellite instability-high colon cancer was the main tumor model in the initial clinical trials demonstrating anti-PD-1 efficacy [63], and MSI-h is the biomarker behind pembrolizumab’s historical site-agnostic approval [64]. Currently, anti-PD-1 blockade is indicated for colorectal tumors that are MSI-h or have a tumor mutational burden (TMB) > 10 [65]. However, because most colorectal tumors are microsatellite stable (MSS), there are ongoing studies on possible markers of susceptibility to immunotherapy in this group of tumors. A recent study proposed an immune-related signature, which included five lncRNAs, capable of predicting prognosis and sensitivity to immunotherapy. Although the low-risk group was enriched for MSI-h tumors, as expected, there were MSI-h tumors in the high-risk group and MSS tumors in the low-risk [66]. Interestingly, ZEB1-AS1 is cited in Li et al. [66] and also mentioned in another manuscript proposing a nine lncRNA-based immune-related signature for CRC [67]. Another study using a machine learning-based data integration approach proposed an immune-derived lncRNA signature that could accurately predict an MSI-h phenotype and divide CRC patients into low- and high-risk groups. The low-risk group characterized an “immune-hot” phenotype, with higher TMB, abundant immune cell infiltration, and high PD-L1 expression [68]. Also, a 14-lncRNA-based classifier acted as a surrogate for TMB in CRC [69].

The role of lncRNAs regulating the PD-1/PD-L1 axis is widely investigated in lung cancer. Immune system evasion of lung carcinoma cells was evaluated by Kathuria et al., who explored the axis NKX2-1-AS1/NKX2-1/PD-L1. NKX2-1 is a transcription factor (also known as thyroid transcription factor-1 or TTF-1), which binds to the CD274 promoter, positively regulating its expression. Using RNA immunoprecipitation followed by qPCR, the authors showed that NKX2-1-AS1 acts as a decoy to NKX2-1, preventing its binding to the CD274 promoter and thus negatively regulating endogenous PD-L1, limiting immune system evasion of lung adenocarcinoma cells [70]. Other axes involving lncRNAs studies in lung cancer cell lines include PSMA3-AS1/miR-17-5p/PD-L1 [71], SChLAP1/AUF1/PD-L1 [72], and SOX2-OT/miR-30d-5p/PDK1 [73]. This later seems to mediate PD-L1 checkpoint regulation through the mTOR signaling pathway, promoting malignant progression and immune escape. Sun et al. described a lncRNA signature consisting of seven lncRNAs associated with tumor immune infiltration in non-small cell lung cancer. Interestingly, the signature was shown to stratify patients’ overall survival irrespective of PD-1 and PD-L1 expression. Therefore, the authors suggest a potential role as an independent predictive biomarker of treatment response to ICI [74]. Another interesting aspect is described by Qu and cols. is that through alternative splicing, PD-L1 can form a lncRNA molecule; this lncRNA was related to lung adenocarcinoma progression by enhancing c-Myc activity and justify deeper studies for investigating PD-L1-lnc depletion in combination with PD-L1 blockade in lung cancer therapy [75].

LncRNA HOXA transcript at the distal tip (HOTTIP) has been shown to promote immune escape in ovarian cancer by enhancing the binding between the transcription factor c-jun and interleukin-6 (IL-6) promoter. Increased IL-6 secretion led to enhanced PD-L1 expression in neutrophils, resulting in immune evasion and inhibited T cell proliferation due to activation of STAT3 pathway [76].

Another study with gastric cancer cells showed that lncRNA urothelial carcinoma associated 1 (UCA1) upregulates PD-L1 expression through sponging miR-214 and 193a, which usually target PD-L1 3′UTR, also contributing to cancer cell migration, invasion and drug resistance in gastric cancer [77]. Similarly, lncRNA LINC00473 upregulates PD-L1 expression in pancreatic cancer through sponging miR-195-5p [78].

An integrative analysis of lncRNAs associated with tumor immune response including over 9000 samples across 32 cancer types was able to define a score (LIMER—lincRNA-based immune response) able to predict immune cell infiltration and prognosis [31]. In the same study, the authors described that lincRNA EPIC1 (epigenetically induced lincRNA-1) interacts with the histone methyltransferase EZH2, leading to the epigenetic silencing of several genes, including MHC-I genes, representing a possible mechanism of tumor immune evasion and, thus, resistance to immunotherapy.

The wider the understanding of the regulation of the PD-1/PD-L1 pathway, the more biomarkers can be discovered, so that the ability to select patients that can benefit from immunotherapy properly is refined. In Figure 2 we organized the data of lncRNAs and the PD-1/PDL-1 pathway in all the discussed cancer types.

### 3.3. lncRNAs Modulated by Treatment with ICI

We also verified data regarding the modulation of lncRNA expression in response to treatment with ICI in cancer, including TNBC. In the literature available up to the date of the research (August 2023), few data are available verifying this biological relationship.

In a study published by Zhang et al. [79], the authors created a molecular signature of genes in response to treatment with ICI using several Pan-Cancer datasets, including BC. Some lncRNAs were found from this list, such as LINC00844, LINC01268, MIR205HG, and TINCR.

Additional studies were not found in the literature using the TNBC as a model. However, we identified in the literature a database called TIGER (http://tiger.canceromics.org/#/home, accessed on 2 August 2023), which has data concerning the modulation of gene expression in response to ICI in other cancer types [80]. Using this database, we searched for the lncRNAs highlighted in [79]. LINC00844 has evidence of increased expression in melanoma in response to anti-PD-1 treatment, and in contrast, TINCR is shown to have decreased expression under the same conditions. These results demonstrate the importance of further investigating the molecular aspects of these lncRNAs in TNBC in response to ICI.

## 4. Tying the Knots: Immunotherapy and TNBC

Considering that around 20% of TNBC tissues are PD-L1 positive and that overall response to PD-1/PD-L1 blockade therapies in that population ranges from 10 to 18.5% [49], while there are studies for several cancer types reporting efficacy of ICI in PD-L1 negative tumors [81,82], it is clear there is a gap in the recognition of best candidates to immunotherapy. Whether ncRNAs can fulfill this gap is yet to be answered, and studies comparing lncRNA signatures of immune infiltration with PD-L1 status and response are lacking. This comparison could help us understand if lncRNAs can be independent predictors of susceptibility to ICI, adding to the current patient selection strategies.

In order to highlight new lncRNAs that could be involved in PD-L1 regulation in TNBC, we conducted a bioinformatic analysis to investigate the role of lncRNAs related to PD-1/PDL-1 pathway in other cancer types (previously discussed in this article) in breast cancer basal-like samples. For this, we used TCGA BRCA cohort data available from the GDC repository that was downloaded via TCGAbiolinks R package (2.24.3 version), comparing lncRNAs expression in basal-like samples compared to the non-tumor breast using limma R package (3.16 version). The results of this analysis are presented in Table 3 below.

The lncRNA MILIP (MAFG-DT) was the most up-regulated in basal-like samples. This lncRNA was poorly investigated in the BC context. In the study conducted by Su and colleagues, MAFG-DT expression was related to BC’s poor prognosis [83]. MAFG-DT was also described to control a ceRNA axis in BC. The axis MAFG-DT/miR-150-5p/MYB is proposed to have an oncogenic role in BC, inducing BC cell line proliferation and migration [84]. The same oncogenic effect was observed in MAFG-DT/miR-574-5p/SOD2 axis [85]; MAFG-DT/miR-339-5p/MMP15 [86], and MAFG-DT/miR-3196/TFAP2A [87]. The lncRNA MAFG-DT was also correlated with BC tamoxifen resistance [88]. Besides the higher expression of MAFG-DT in basal-like samples when compared to normal breast, we did not observe a significant positive expression correlation value between this lncRNA and PD-L1 (r = −0.25; *p* < 0.05).

So, in order to find a lncRNA that is both up-regulated in basal-like samples and positively correlated to PD-L1, we performed a Pearson’s correlation analysis in the following lncRNAs: NARF-AS1, WASIR2, OGFRP1, PART1, UCA1, NKX2-1-AS1, and HCP5. We found a significant positive correlation analyzing HCP5 and PD-L1 in basal-like samples (r = 0.72; *p* < 0.01). HCP5 has already been related to TNBC progression by regulating miR-219a-5p/BIRC3 [89]; also, its high expression was related to cisplatin resistance in the TNBC cell line [90] and associated with BC poor prognosis [91]. In addition, HCP5 was associated with PD-L1 regulation by sponging miR-150-5p [92]. The role of HCP5 in TNBC and PD-L1 regulation should be better investigated and explored.

Besides that, we investigated the lncRNAs presented in Table 3, considering the immune profile from the same TCGA basal-like data. For that, we used the immune subtypes published by Thorsson and colleagues [26]. In this study, the authors proposed six immune signatures using TCGA samples. Two subtypes stand out due to the number of samples in each group: C1 (wound healing), which has high proliferation rates and high expression of angiogenic genes, and C2 (IFN-γ dominant), which has the highest M1/M2 macrophage polarization. We compared the expression of the 11 lncRNAs between C1 x C2 immune subtypes and the results obtained from this analysis. The result generated by our differential expression analysis is represented in the heatmap in Figure 3. The nomenclature of the immune subtypes used was taken from Thorsson and colleagues [26].

Interestingly, only two lncRNAs had significant *p* < 0.05 differential expression values: HCP5 and UCA1. In our analysis, HCP5 and UCA1 are up-regulated in the C2 subtype. HCP5 has already been discussed here as a potential lncRNA impacting BC immunotherapy, since it is overexpressed in basal-like subtypes, and exhibits a high correlation with the PD-L1 gene. The C2 subtype, which has up-regulation of HCP5 and UCA1 is an immune type that has a high M1/M2 macrophage polarization, a strong CD8 signal, and the greatest TCR diversity [26]. In fact, a diverse repertoire of T cells directed against a small number of cancer-specific targets is key to the success of immunotherapy [93].

HCP5 is located centromeric of the HLA-B gene and between the MICA and MICB genes within the major histocompatibility complex (MHC) class I region and research found that this molecule is involved in adaptive and innate immune responses [94]. In BC, HCP5 a ceRNA network, BTN3A1-hsa-miR-20b-HCP5 is discussed to be the potential interaction mechanism between BC and immune cells [95]. The role of this lncRNA in immunotherapy seems to be an interconnected pathway that should be further investigated.

The UCA1 role in immune pathways in BC has not been investigated yet. Nonetheless, this lncRNA was highlighted in immune escape pathways in other cancer types, such as endometrial [96] and gastric [77]. In line with what was discussed for HCP5, it should be better explored in breast cancer.

## 5. Final Considerations

In this review, we highlighted lncRNAs involved in PD-L1 pathway regulation in several cancer types, focusing on TNBC for which pembrolizumab was recently approved. We also performed an in-silico analysis bringing together lncRNAs associated with PD-L1 in other cancer types and evaluated its expression and correlation on TNBC patients.

This information can highlight lncRNAs to be better explored in the TNBC context, especially considering immunotherapy eligibility. These findings may guide future research using these lncRNA as targets, also thinking about pathways related to immune escape.

## 6. Conclusions

Based on what was discussed in this work, lncRNAs can significantly contribute as biomarkers for screening patients eligible for anti-PD-1 antibody treatment. These molecules may be better explored to predict ICI treatment responsiveness, considering that the expression of PD-1/PDL-1 proteins does not reflect, in some cases, responsiveness to treatment.

## Figures and Tables

**Figure 1 cancers-15-04682-f001:**
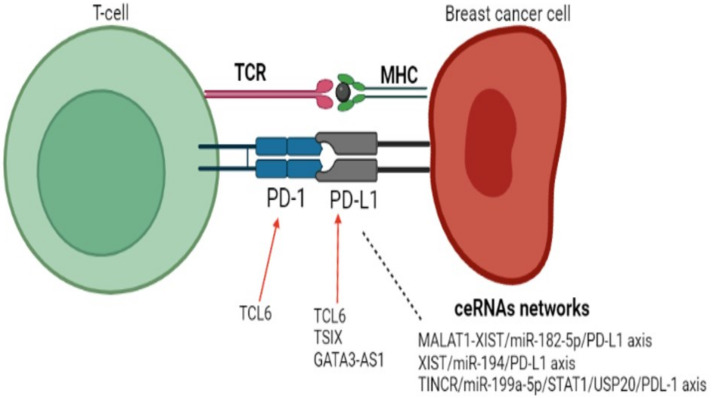
lncRNAs in BC related to the PD-1/PDL-1 pathway. Red arrow: expression up-regulation.

**Figure 2 cancers-15-04682-f002:**
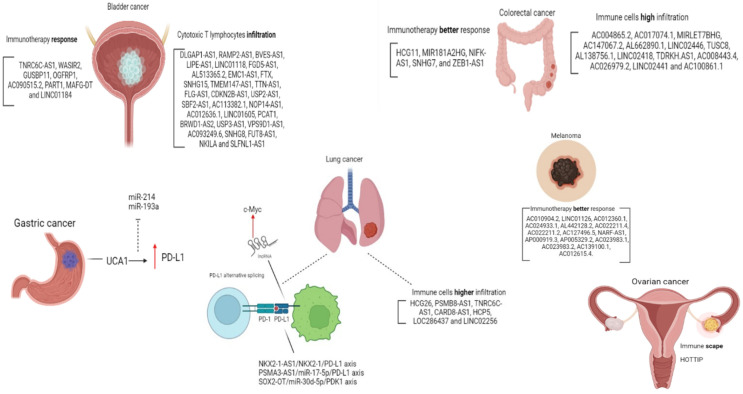
lncRNAs in different cancer types related to the PD-1/PDL-1 pathway. Red arrow: expression up-regulation. Can ICI, including Pembrolizumab, alter lncRNAs expression in cancer.

**Figure 3 cancers-15-04682-f003:**
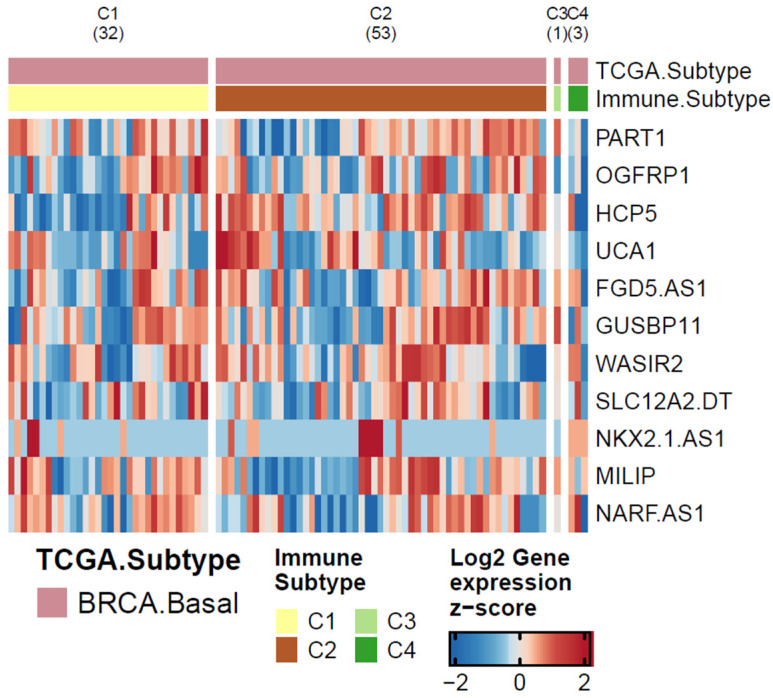
Heatmap representation of lncRNA’s expression previously cited in other cancer types as being related with PD-1/PDL-1 pathway. Immune subtypes were extracted for Thorsson et al. considering only breast samples classified as basal-like.

**Table 1 cancers-15-04682-t001:** Triple-negative breast cancer classifications including subtypes related to tumor immune response.

Subtype	Clinical Significance	Reference
(a) BL1, BL2, IM, M, MSL, LAR	The described IM subtype was related with better prognosis	[5]
(b) T, B, E and D (according to stroma axes)	Signature classified patients according to outcomes	[15]
(c) C1, C2 and C3	C2 and C3 were sensitive to therapy combating immunosuppression	[16]
(d) MC1, MC2, MC3, MC4, MC5 and MC6	High level of CD8+ and CD4+ immune signatures in MC6 subtype.	[18]
(e) Immunity H, Immunity M, Immunity L	Immunity H subtype was related with better prognosis as it shows more immune cells expression	[19]
(f) LAR, basal, claudin-low, claudin-high and two immune subtypes	Claudin-h and immune-positive subtypes have better prognosis	[22]
(g) Epi-CL-A, Epi-CL-B, Epi-CLC, Epi-CL-D	Epi-CL-D showed a positive regulation of T lymphocyte cytotoxicity, enriched response to interferon-beta and antigen processing	[23]
(h) Immune phenotype A and B	Phenotype A exhibited an enrichment of immune-related pathways	[24]
(i) ImA, ImB and ImC	Systemic inflammation parameters are informative markers	[25]
(j) C1, C2, C3, C4, C5 and C6	C2 immune subtype is related with immune activation	[26]

Different classifications including immune-related subtypes have been proposed to breast cancer, including TNBC. (a) BL1: basal-like 1; BL2: basal-like 2; IM: immunomodulatory; M: mesenchymal; MSL: mesenchymal stem-like; LAR: luminal androgen receptor. IM subtype is enriched for gene ontologies in immune cell processes and immune signal transduction pathways. (b) T: enriched for T cells; B: enriched for B cells; E: enriched for epithelial markers; D: enriched for desmoplasia. These four stromal axes interact, and though the T predictor was one of the best predictors, the prognostic capacity of the B, T, and E scores are governed by the D score. (c) C1: molecular apocrine; C2: basal-like-enriched with pro-tumorigenic immune response; C3: basal-like-enriched with adaptive immune response. C2 presents with high aggressiveness, while C3 exhibits immune checkpoint upregulation. (d) MC1, MC2, MC3, MC4, MC5 and MC6: molecular complexity 1 to 6. Immune gene enrichment-based classification, with activated CD4+ and CD8+ immune signatures, was found to be enriched in MC6 and low in MC1. (e) Immunity_H: High infiltration of immune cells and antitumor activity, and better survival rate compared to other subtypes; Immunity_M: infiltration of immune cells and intermediate antitumor activity, with an intermediate survival rate compared to the other subtypes; Immunity_L: Low infiltration of immune cells and high oncogenic activity, with worse survival rate compared to other subtypes. (f) LAR: luminal androgen receptor. Immune-positive subtypes have better prognosis. (g) Epi-CL-A, Epi-CL-B, Epi-CL-C and Epi-CL-D. Epi-CLD showed enriched response to interferon-beta, antigen processing and presentation, and positive regulation of T cell-mediated cytotoxicity and presented with improved outcomes. (h) In immune phenotype A, immune-related pathways were significantly enriched, and a higher level of immune checkpoint molecules, presenting with a better overall survival. (i) ImA, ImB, and ImC: immune-cluster A to C. Immune A is an immune-active subtype associated with favorable prognosis. (j) C1, C2, C3, C4, C5 and C6: immune subtypes 1 to 6. The classification is not specific for breast cancer; C3 had the best prognosis, while C2 and C1 had less favorable outcomes despite having a substantial immune component.

**Table 2 cancers-15-04682-t002:** Immune-related lncRNA prognosis signatures in breast cancer.

lncRNAs	Measured Outcome	Reference
OTUD6B-AS1, AL122010.1, AC136475.2, AL161646.1, AC245297.3, LINC00578, LINC01871, AP000442.2	Overall survival	[51]
MAPT-IT1, SLC26A4-AS1, VPS9D1-AS1, PCAT18, LINC01234, SPATA41, LINC01215	Overall survival	[53]
AC116366.1, AC244502.1, AC100810.1, MIAT, AC093297.2, AL356417.2	Overall survival	[52]
LINC01010, AP005131.6, AC004847.1, AL591686.1, LINC00668, LINC02418, AL356515.1, AC027514.1, AL772337.1, AL161646.2, AC243773.2	Overall survival and immune cell infiltration	[54]

**Table 3 cancers-15-04682-t003:** LncRNAs differentially expressed in basal-like x normal breast tissue samples using TCGA BRCA cohort data.

lncRNA	logFC	*p*-Value
NARF-AS1	0.38	1.64 × 10^−33^
WASIR2	0.57	6.54 × 10^−10^
GUSBP11	−0.17	0.003
OGFRP1	0.55	6.28 × 10^−31^
PART1	0.87	1.40 × 10^−10^
MILIP (MAFG-DT)	1.28	9.20 × 10^−40^
LINC01184	−0.44	5.09 × 10^−11^
UCA1	0.87	8.50 × 10^−13^
FGD5-AS1	−0.21	0.004
NKX2-1-AS1	0.36	5.70 × 10^−26^
HCP5	0.68	1.48 × 10^−5^

LncRNAs that were differentially expressed in basal-like x normal breast comparison considering adjusted *p*-value < 0.05 below in the Wald test.

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
