# Peer review of "PD-1/PD-L1 Inhibitors Response in Triple-Negative Breast Cancer: Can Long Noncoding RNAs Be Associated?"

_cancers, 2023, doi:10.3390/cancers15194682_

Round 1

Reviewer 1 Report (Previous Reviewer 1)

In this review article, the authors investigated the potential role of long non-coding RNAs (lncRNAs) on the regulation of the PD-1/PD-L1 pathway, as potential biomarkers. They also conducted a bioinformatic analysis to investigate lncRNAs already investigated in PD-1/PDL-1 pathways in other cancer types considering TNBC molecular context. The authors have how a lot of evidence for the potential role of lncRNAs in tumor prognosis and as potential diagnostic markers especially in TNBCs.

The authors have shown a lot of evidence in support of they hypothesis but felt that some parts of the paper were more complicated than needed and sometimes diverted away from the breast cancers which is your primary focus. At times it read more like a collection of results and loses a bit of the story, but I agree that the potential role of lncRNAs need further investigation for the treatment of cancer. I do not have any major concerns with the paper and only recommend a few minor corrections.

The English is very good and not difficult to understand. This version of the paper is easier to follow but I only have a couple of minor recommendations with respect to the English in the other comments. I do still find that the paper would benefit from a bit more focus rather than the large number of lncRNA discussed throughout the paper.

 Lines 241/242 – “immune checkpoint inhibitor” is a different font size.

 Section on other cancers is a bit long and is a bit distracting from the focus on TNBC

Not sure there is enough discussion on the drug Pembrolizumab for it to be included in the title as no work was done by the authors on it and several others medications are also noted.

Author Response

Revisor 1

“In this review article, the authors investigated the potential role of long non-coding RNAs (lncRNAs) on the regulation of the PD-1/PD-L1 pathway, as potential biomarkers. They also conducted a bioinformatic analysis to investigate lncRNAs already investigated in PD-1/PDL-1 pathways in other cancer types considering TNBC molecular context. The authors have how a lot of evidence for the potential role of lncRNAs in tumor prognosis and as potential diagnostic markers especially in TNBCs.

The authors have shown a lot of evidence in support of they hypothesis but felt that some parts of the paper were more complicated than needed and sometimes diverted away from the breast cancers which is your primary focus. At times it read more like a collection of results and loses a bit of the story, but I agree that the potential role of lncRNAs need further investigation for the treatment of cancer. I do not have any major concerns with the paper and only recommend a few minor corrections.

 The English is very good and not difficult to understand. This version of the paper is easier to follow but I only have a couple of minor recommendations with respect to the English in the other comments. I do still find that the paper would benefit from a bit more focus rather than the large number of lncRNA discussed throughout the paper.

Section on other cancers is a bit long and is a bit distracting from the focus on TNBC”

Answer:

Dear reviewer,

We appreciate the comments made regarding our manuscript.

We cited the relationship of lncRNAs associated with the PD-1/PD-L1 pathway in other tumors to have hypotheses and to investigate better these molecules in the context of triple-negative breast cancer. But we agree that the number of lncRNA and mechanisms citations in some parts may have made the paper more complicated. 

To try to make the manuscript more focused and with fewer lncRNA citations, we reviewed all the text, especially item ‘b) lncRNAs and PD-1/PDL-1 axis in other cancer types’, removing that number of lncRNAs cited in the text and leaving this information only in Figure 2.

“ Lines 241/242 – “immune checkpoint inhibitor” is a different font size.”

Answer:

Thank you for this point, we corrected.

 “Not sure there is enough discussion on the drug Pembrolizumab for it to be included in the title as no work was done by the authors on it and several others medications are also noted.”

Answer:

We completely agree. Although the drug Pembrolizumab is in the spotlight due to the recent FDA authorization, the article is more focused on the pathway and we have changed the title: “PD-1/PD-L1 inhibitors in triple-negative breast cancer: can long noncoding RNAs emerge as biomarkers?”

Reviewer 2 Report (Previous Reviewer 2)

there is no discussion of a direct link between Pembrolizumab and a lncRNA

moderate

Author Response

“there is no discussion of a direct link between Pembrolizumab and a lncRNA”.

Dear reviewer,

We appreciate the comments made regarding our manuscript. Certainly, it was not clear in the previously submitted manuscript that the objective of the work is to raise molecules of lncRNAs that can act as biomarkers for indication of immunotherapy in TNBC. Therefore, our objective is to bring out the potential use of lncRNAs to aid in the patient’s selection. In this way, we restructured some parts of the article in order to clarify the objectives proposed by us in this review. Highlighted text can best be seen underlined in yellow.

Additionally, although the drug Pembrolizumab is in the spotlight due to the recent FDA authorization, the article is more focused on the pathway. So, we have changed the title: “PD-1/PD-L1 inhibitors in triple-negative breast cancer: can long noncoding RNAs emerge as biomarkers?”

Further, we also verified data regarding the modulation of lncRNA expression in response to treatment with ICI in cancer, including TNBC. To highlighted that we also researched for this association (although few data was found), we included the item topic “lncRNAs modulated by treatment with ICI”

English review of the entire manuscript was done.

Reviewer 3 Report (Previous Reviewer 3)

The authors made some changes in this revised version.  I would consider that these changes are an improvement to the clarity of some aspects. However, there are some wording issues on the language side in those additions to the original version. The authors need to check through the new parts and make corrections.

For examples:

1.     Line 13, need to make change to “indicate the treatment”.

2.     Line 28, need to make change to “we evidence 2 lncRNAs”.

3.     Line 152, “the expression of those PD-L1 on tumor surface” may be changed into “PD-L1 expression on tumor cell surface”.

4.     Line 215, “established” may changed into “reported”. 

5.     Line 266, the sentence looks awkward.

6.     In Conclusions, sentences need to be changed. In Line 448, “with the aim of improving the effectiveness of the treatment” is not a right description. These biomarkers can predict ICI treatment responsiveness, but this is not equal to actually improving treatment effectiveness. 

The new additions have much more issues than the original submission on the language side.

Author Response

“The authors made some changes in this revised version.  I would consider that these changes are an improvement to the clarity of some aspects. However, there are some wording issues on the language side in those additions to the original version. The authors need to check through the new parts and make corrections.

For examples:

  1. Line 13, need to make change to “indicate the treatment”.
  2. Line 28, need to make change to “we evidence 2 lncRNAs”.
  3. Line 152, “the expression of those PD-L1 on tumor surface” may be changed into “PD-L1 expression on tumor cell surface”.
  4. Line 215, “established” may changed into “reported”. 
  5. Line 266, the sentence looks awkward.
  6. In Conclusions, sentences need to be changed. In Line 448, “with the aim of improving the effectiveness of the treatment” is not a right description. These biomarkers can predict ICI treatment responsiveness, but this is not equal to actually improving treatment effectiveness. “

Dear reviewer,

We appreciate the comments made regarding our manuscript. We corrected all pointed examples, including the change of line 266 and conclusion sentence suggestion. Additionally, English review of the entire manuscript was done.

Round 2

Reviewer 2 Report (Previous Reviewer 2)

thank you for your edits. 

for the title, it is not clear how lncRNA can be biomarkers of PDL-1 inhibitors unless you are follow patients treated with PDL-1 inhibitors, and monitoring clinical endpoints.

please remove the biomarkers statement in the title

.......

Author Response

Thank you very much for the suggestion. We changed the title: "PD-1/PD-L1 inhibitors response in triple-negative breast cancer: Can long noncoding RNAs be associated?"

This manuscript is a resubmission of an earlier submission. The following is a list of the peer review reports and author responses from that submission.

Round 1

Reviewer 1 Report

In this review article, the authors investigated the potential role of long non-coding RNAs (lncRNAs) on the regulation of the PD-1/PD-L1 pathway, as potential biomarkers. They also conducted a bioinformatic analysis to investigate lncRNAs already investigated in PD-1/PDL-1 pathways in other cancer types considering TNBC molecular context. The authors have how a lot of evidence for the potential role of lncRNAs in tumor prognosis and as potential diagnostic markers especially in TNBCs.

The abstract does not really include a concluding statement of findings.

Table 1 does not have the a,b,c,d.. etc labeling which is in the legend for this table.

Line 122. There should be a period at the end of the sentence.

Line 132 – I think “and” should be changed to “in which” or “where” so it reads ; “Immune evasion is one of the cancer hallmarks which can be targeted for treatment, including the PD-1/PD-L1 pathway, where lncRNAs have also been previously described acting in such biological processes [28].”

Line 139 – I think “later” should be “latter” which indicates the last one in the list.

The English is very good and not difficult to understand. I only have a couple of minor recommendations with respect to the English in the other comments. I do not recommend a restructure of the paper but hope authors will have a bit more focus in future publications.

The authors have shown a lot of evidence in support of they hypothesis but felt that some parts of the paper were more complicated than needed and sometimes diverted away from the breast cancers which is your primary focus. At times it read more like a collection of results and loses a bit of the story, but I agree that the potential role of lncRNAs need further investigation for the treatment of cancer. I do not have any major concerns with the paper.

Author Response

We thank the reviewer for critically reviewing the article. The improvements will contribute significantly to the enrichment of our article. All the modification in the manuscript are highlighted in yellow for better visualization.

Reviewer 2 Report

the authors present an eloquent manuscript on " Immune checkpoint inhibitors in triple-negative breast cancer: can long noncoding RNAs emerge as biomarkers"

The review is certainly short in scope, with broad concdpts being introduced with little to no explanation. 

first, there are not really ten different classifications of TNBC.  

second, the lncRNAs in table 2 are not sufficient. 

third, ceRNA is described in a figure legend which little explanation in the text. 

fourth, there is much mention of PD-1.  and perhaps the review should focus on that subtopic. '

figure 3 looks copied from another figure without proper citation. 

please edit

Author Response

We thank the reviewer for critically reviewing the article. The improvements will contribute significantly to the enrichment of our article. All the modifications in manuscript are highlighted in yellow for better visualization.

Reviewer 3 Report

The approval of immune checkpoint inhibitors by FDA as a cancer immunotherapy represents the most conspicuous advance of recent years in cancer research. On the other hand, only a portion of patients of diverse cancer types respond well to immune checkpoint inhibitors. This review article by Mathias et al. discusses the feasibility of using long noncoding RNAs as biomarkers to determine responsiveness of triple- negative breast cancer to immune checkpoint inhibitors. The authors did a wonderful job in curating and summarizing related literature. In addition, the authors carried out a bioinformatic analysis of lncRNAs already discussed in other cancer types in basal-like breast cancer samples. These provide an objective analysis that may shed light on future investigation of lncRNA as biomarkers for selection of triple-negative breast cancer patients to be treated with immune checkpoint inhibitors. I would recommend acceptance of this manuscript for publication. Aslo, I would like to see additional discussion in the manuscript on how lncRNA expression is regulated by cellular factors in the context of triple negative breast cancer. 

Minor points:

1.  Line 114, “PD-/PD-L1” should be “PD-1/PD-L1”.

2.   Find a description that clarifies “basal-like x normal breast” in Table.

3. Fig. 1 and Fig. 2 look fuzzy.

Author Response

We thank the reviewer for critically reviewing the article. The improvements will contribute significantly to the enrichment of our article. All the modification in the manuscript are highlighted in yellow.

Round 2

Reviewer 2 Report

The authors made some attempts to revise the manuscript.

Unfortunately the manuscript is still not well developed. 

The topic of PD-1 Pembrolizumab and lncRNA development are not well linked.   There is little evidence Pembrolizumab alters lncRNA expression levels. 

This topic is not discussed until pages 6 or 7 of the manuscript

Pembrolizumab

.

Author Response

Dear reviewer,

We appreciate the comments made regarding our manuscript. Certainly, it was not clear in the previously submitted text, is that the objective of the work is to raise molecules of lncRNAs that can act as biomarkers for indication of immunotherapy in TNBC. In the opening pages of the manuscript, we focused on discussing the immune aspects related to TN BC tumors, which are mostly classified as immunogenic. However, what has  been observed is that t he presence of PD-1 in the tumor sample is not related to 100% success in treatment with the drug (Pembrolizumab). This information can be accessed in the article in the paragraph: “Within this context, the expression of those PD- L1 on tumor surface is currently used as one of the biomarkers to assess eligibility for immunotherapy, along with microsatellite instability (MSI) and tumor mutational burden (TMB) – of note, all these markers are rarely concordant, but each of them suffices as a marker [40]”.

In addition, when developing  information on immunotherapy in TNBC :”Considering that around 20% of TNBC tissues are PD-L1 positive and that overall response to PD-1/PD-L1 blockade therapies in that population ranges from 10 to 18.5% [49], while there are studies for several cancer types reporting efficacy of ICI in PD-L1 negative tumors [79,80], it is clear there is a gap in the recognition of best candidates for immunotherapy”. reinforces the idea of using new molecules as biomarkers.

Therefore, our objective is to bring out the potential use of lncRNAs to aid in the patient’s selection. In this way, we restructured some parts of the article in order to clarify the objectives proposed by us in this review. Highlighted text can best be seen underlined in yellow.

Round 3

Reviewer 2 Report

Unfortunately the manuscript is still not well developed. 

The topic of PD-1 Pembrolizumab and lncRNA development are not well linked.   There is little evidence Pembrolizumab alters lncRNA expression levels. 

The table of lncRNA in TNC reference basal-like TNBC, not an immune-deficient or altered tumor type 

N/A